# Dynamics of a Turbine Blade with an Under-Platform Damper Considering the Bladed Disc's Rotation

**Shangwen He [1], Wenzhen Jia [1], Zhaorui Yang [1,*], Bingbing He [2] and Jun Zhao [1]**

[1] School of Mechanics and Safety Engineering, Zhengzhou University, Zhengzhou 450001, China; hsw2013@zzu.edu.cn (S.H.); jiawenzhen1993@163.com (W.J.); zhaoj@zzu.edu.cn (J.Z.)

[2] College of Mechanical and Electrical Engineering, Shaanxi University of Science and Technology, Xi'an 710021, China; hebb714@gmail.com

\* Correspondence: zryang@zzu.edu.cn; Tel.: +86-1856-760-1945



**Featured Application: This study supplies a clear thinking to improve the dynamic model of turbine blade with under-platform damper. With this thinking, deeper study can be done to make the damper design more scientific.**

**Abstract:** To simplify the dynamic model, it is generally assumed that the bladed disc is stationary in current studies on the dynamics of the turbine blade with an under-platform damper. With this assumption, convective inertial force in tangential direction and Coriolis inertial force are not be considered in the dynamic model and its equations. To make the dynamic model and relevant analysis more scientific, an approximation method has been developed with the theory of compositive motion. With this method, the response of the blade relative to the rotating bladed disc could be calculated, and the influence of the bladed disc's rotation is considered in this paper. Considering the bladed disc from startup to steady-state, the dynamic characteristics of the system were studied. The influence of damper mass, damper vibration stiffness, and external excitation amplitude on the vibration reduction characteristics of the system were obtained. A method for determining the time when the system reaches steady-state vibration was proposed with the normalized cross-correlation function (NCCF) and the bisection method. The simulation results show that the bladed disc's rotation has an obvious influence on the dynamic characteristics of the system, and some new conclusions were obtained.

**Keywords:** under-platform damper; bladed disc's rotation; compositive motion; relative displacement; dynamical characteristic

## 1. Introduction

High-cycle fatigue (HCF) failure of the turbine blades of aero-engines caused by high vibrational stresses is one of the main causes of aero-engine incidents. Due to its insensitivity to temperature and its simple structure, the under-platform damper was widely used to reduce the vibration of the aero-engine turbine blades [1]. To predict the response of the turbine blade with an under-platform damper more and more accurately, recently, there have been quite a lot of developments in the calculations and analyses of under-platform dry friction damper. These studies are mainly about structural dynamic model, dry friction contact model, and methods for solving response of the nonlinear system.

Menq [2] and Sanliturk [3] studied the friction contact and the effect on the vibration reduction of the two-dimensional friction motion. Xia [4] proposed a model for investigating the stick-slip motion caused by dry friction of a two-dimensional oscillator under arbitrary excitations and provided a numerical approach to investigate the system with the Coulomb friction law. Shan and Zhu [5,6] used the numerical tracking method to analyze the complex motion and studied the dynamic response

of the plate blade with an under-platform damper. Ma et al. established a dynamic model of rotating shrouded blades considering the effects of the centrifugal stiffening and spin softening of the blade [7]. He B and Ouyang H studied the forced vibration response of a turbine blade with a new kind of under-platform damper, in which the vertical motion of the damper leads to time-varying contact forces and can cause horizontal stick-slip motion [8]. For understanding the actual dynamics of the blade–damper interaction, a novel experimental test rig was developed to extensively investigate the damper's dynamic behavior [9,10]. Umer and Botto [11] explored the contact forces and relative displacement between the damper–blade contact interface with an experimental study for the first time. Liao and Li [12] proposed a two-dimensional friction ball/plate model and established a dynamics model of the rotor with elastic support/dry friction dampers.

Qi and Gao [13,14] established a one-dimensional macro-micro slip friction model to analyze the dynamic characteristics of the damper system, and compared it with the results of the finite element method. The phenomenological macro-slip of dry friction modeling was described in mathematical form by two approaches, and both approaches were illustrated using different acceleration excitations to describe the differences between them [15]. A model was proposed to characterize friction contact of non-spherical contact geometries obeying the Coulomb friction law with constant friction coefficient and constant normal load and the dissipated energies were obtained for different contact geometries [16]. A decrease in vibrational amplitudes was explained by changes in boundary conditions induced by a stick/slip behavior, and the contribution of respective energy dissipation and change of contact state on peak levels was shown [17]. He and Ren [18] studied the reducing vibrational characteristics of the blade by the two-dimensional friction model and finite element model. A purposely developed contact model was tuned on a single-contact test and then included in the numerical model of a curved-flat damper to simulate its cylindrical interface [19]. A microslip model was developed for analyzing the damping effect of under-platform dampers for turbine blades, but the inertia and rotating effects of the damper were ignored for simplicity [20].

Wang and Zhang [21] studied the free vibration and forced vibration of a dry friction oscillator, which was composed of the Iwan model and a mass by harmonic balance method. The multi-harmonic balance method was used to analyze the periodic vibrations of the damper system and to investigate the steady-state solutions of the nonlinear system [22]. A method to predict the nonlinear steady-state response of a complex structure was described, and two differential forms of friction force were given to solve the tangential force of the blades with under-platform dampers accurately [23]. The vertical contact forces and the resultant friction forces acted as moving loads, and the finite element method and the modal superposition method were used to obtain the numerical modes and to solve the dynamic response of the dry friction dampers [24]. Yu and Xu studied the properties of cubic nonlinear systems with dry friction damping and an approximate method was used to get the frequency-response function [25].

In the most studies, it is generally assumed that the bladed disc is stationary, to simplify the dynamic model in the design and analysis of the blade. Bladed disc's rotation is considered in some studies from the aspect of the centrifugal stiffening of the blade. There are few studies about the improvement of the dynamic model considering the bladed disc's rotation. Besides, the study of the dynamic characteristics of the whole process from startup to steady-state has not been included.

In engineering practice, the blades are set up in a circle around the disc, and the under-platform damper can be installed between two adjacent blades. The whole structure can be considered to be in cyclic symmetry. If the normal pressure between contact surfaces is supposed to be distributed equally between two adjacent blades, then the structural model of under-platform damper, as used in this paper, can be described as in Figure 1, where the *xyz* orthogonal coordinate system (called the moving coordinate system) is defined in accordance with the axial (*x*, along to the angular velocity direction of the blade), tangential (*y*), and radial (*z*) directions. The coordinate system is attached to the bladed disc and rotates with it. A static coordinate system fixed to the ground is defined.

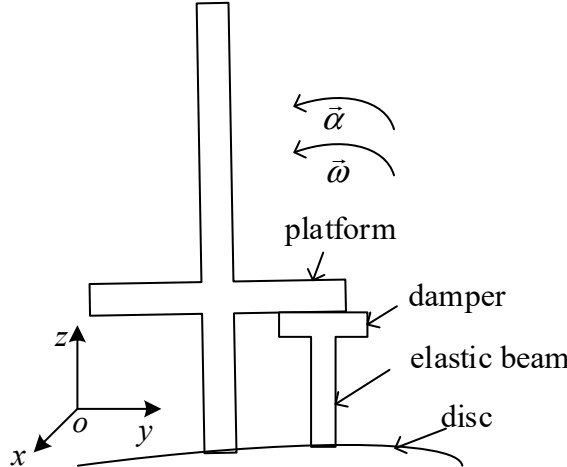

**Figure 1.** Structure of blade with under-platform damper.

When the rotation of the bladed disc is considered, firstly the vibration stress of the blade mainly depends on its relative displacement (response) to the bladed disc, and the relative displacement should be studied instead of absolute displacement. Secondly, at the rotating state the variation of the convective inertial force and Coriolis inertial force leads to the change of normal pressure and tangential force, which has a significant influence on the damping effects of the damper and the dynamic characteristics of the blade. To study the influence of bladed disc's rotation and improve the accuracy of analysis of the damper, an approximation method for the dynamic response of the blade relative to the bladed disc (moving coordinate system) has been proposed in this paper by combining the theories of compositive motion and dynamics. Compositive motion describes the motion of a moving body relative to different coordinate systems. The response before steady-state could be defined as the transient response. With this method, the convective inertial force and Coriolis inertial force are considered in dynamic equations; the properties of the system solution are derived; and the vibration-damping law of the steady-state response and the transient response of the blade are studied. Some new conclusions about the damping characteristics of the turbine blade with an under-platform damper were obtained, which will be useful in the damper design in engineering practice.

## 2. Mechanical Model and Dynamics Equations

### 2.1. Blade-Damper Model

In accordance with Figure 1, the vibration of the first bending mode of the blade in $y$ direction is considered in this paper. The vibrations of the blade in the $x$ and $z$ directions will be neglected, as they are higher order modes and very small relative to the displacement in the $y$ direction. The blade is taken as the moving body; the bladed disc is the moving coordinate system. The convective motion, the motion of the moving coordinate system relative to the static coordinate system, is the bladed disc's rotation about a fixed axis. The motions of the blade and damper relative to the bladed disc can be approximated as the linear motion along the $y$ direction. In Figure 1, $\vec{\omega}$ is the angular velocity of the bladed disc; $\vec{\alpha}$ $(\vec{\alpha} = d\vec{\omega}/dt = \dot{\vec{\omega}})$ is the angular acceleration. The absolute motion of the blade consists of rigid body rotation and vibration relative to the bladed disc. It is assumed that the rigid damper is mounted on an elastic beam and that there is no friction between the damper and the elastic beam. When the bladed disc is rotating, the damper is pressed on the platform by the normal pressure which contains the centrifugal force, Coriolis inertia force and the component of the damper's gravity. The damper and the platform are not separated during the whole process. The sizes of the damper and the blade are ignored for approximate calculation. Based on the above assumption, the structure of under-platform damper (Figure 1) is simplified as a spring-mass model, which is shown in Figure 2.

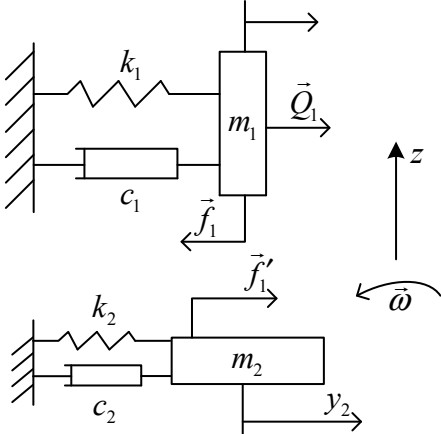

**Figure 2.** Dynamic model of a blade with an under-platform damper.

Two local coordinate systems parallel to the $oxyz$ coordinate system are defined at the platform and the damper, and are stationary relative to the bladed disc. Displacement of the blade relative to the corresponding local coordinate system is $y_1$, while relative displacement of the damper is $y_2$, both of which are in the $y$ direction. The equivalent mass of the blade is $m_1$. The total equivalent mass of the damper and the elastic beam given by the energy method is $m_2$. The vibrational stiffnesses of the blade and damper in the $y$ direction are $k_1$ and $k_2$. The linear damping coefficients of the blade and damper in the $y$ direction are $c_1$ and $c_2$. $\vec{Q}_1$ is aerodynamic excitation force. The dry friction force between the damper and the platform is $\vec{f}_1$, and its reaction force is $\vec{f}_1{}'$.($\vec{f}_1 = -\vec{f}_1{}'$, in $y$ direction).

### 2.2. The Dynamics Equations of Blade-Coupled Vibration

In the $yoz$ plane, the sizes of the damper and the platform are neglected when calculating the convective acceleration. In accordance with Figure 2, the dynamic equation of the system in vector form is established with the theories of compositive motion, which can be written as:

$$\begin{cases} m_1\left(\vec{a}_{1r} + \vec{a}_{1e} + \vec{a}_{1k}\right) + c_1\vec{v}_{1r} + \vec{F}_{K1} = -m_1\vec{g} + \vec{Q}_1 + \vec{f}_1 + \vec{N} \\ m_2\left(\vec{a}_{2r} + \vec{a}_{2e} + \vec{a}_{2k}\right) + c_2\vec{v}_{2r} + \vec{F}_{K2} = -m_2\vec{g} + \vec{f}_1{}' - \vec{N} \end{cases} \tag{1}$$

where $m_1\vec{g}$ and $m_2\vec{g}$ are gravity vectors, which can be ignored, as they are very small in comparison with the other forces. The relative acceleration vectors of $m_1$ and $m_2$ in the corresponding local coordinate systems are $\vec{a}_{1r}$ and $\vec{a}_{2r}$, respectively (in $y$ direction). The corresponding convective acceleration vectors are $\vec{a}_{1e}$ and $\vec{a}_{2e}$. (When $\vec{\omega}$ is constant, the convective acceleration is along the $z$ direction. When $\vec{\omega}$ changes, the tangential component of the convective acceleration is along $y$ direction, and the normal component of the convective acceleration is along the $z$ direction.) The corresponding Coriolis acceleration vectors are $\vec{a}_{1k}$ and $\vec{a}_{2k}$. (The direction of $\vec{\omega}$ is perpendicular to the $yoz$ plane; according to $\vec{a}_k = 2\vec{\omega} \times \vec{v}_r$, the Coriolis acceleration is along the $z$ direction.) The corresponding relative velocity vectors are $\vec{v}_{1r}$ and $\vec{v}_{2r}$(in $y$ direction). $\vec{F}_{K1}$ and $\vec{F}_{K2}$ are the corresponding stiffness forces vectors in $y$ direction. $\vec{Q}_1$ is the aerodynamic excitation forces' vector in $y$ direction. $\vec{N}$ is the normal pressure vector between the blade and the damper in $z$ direction. Ignoring $m_1\vec{g}$ and $m_2\vec{g}$, the dynamic Equation (1) is projected to the $y$ direction, and the scalar form of it could be written as

$$\begin{cases} m_1\ddot{y}_1 + c_1\dot{y}_1 + k_1y_1 = Q_1 - f_1 \\ m_2\ddot{y}_2 + c_2\dot{y}_2 + k_2y_2 = f_1{}' = f_1 \end{cases} \tag{2}$$

$$\begin{cases} m_1(\ddot{y}_1 + a_{1e\tau}) + c_1\dot{y}_1 + k_1 y_1 = Q_1 - f_1 \\ m_2(\ddot{y}_2 + a_{2e\tau}) + c_2\dot{y}_2 + k_2 y_2 = f_1' = f_1 \end{cases} \quad (3)$$

The dynamic equation of the system in $y$ direction is Equation (2) when the rotation speed of the bladed disc is constant, with the rotation speed varying the equation shown as Equation (3). The response characteristics of the blade vibration relative to the bladed disc can be studied at working speed (uniform rotation) and in start-stop condition (non-uniform rotation). In Equation (3), $a_{1e\tau}$ and $a_{2e\tau}$ exist with the bladed disc's rotation considered, and the tangential component of the convective acceleration is described as

$$\begin{cases} a_{1e\tau} = -l_1\alpha = -l_1\dot{\omega} \\ a_{2e\tau} = -l_2\alpha = -l_2\dot{\omega} \end{cases} \quad (4)$$

where $l_1$ and $l_2$ are, respectively, the rotation radius of the platform and the damper rotating around the center of the bladed disc, and could be supposed to be constant values during the motion. The scalar quantity of angular acceleration and angular velocity are $\alpha$ and $\omega$ respectively; the counterclockwise direction is taken as their positive direction.

### 2.3. Normal Pressure and Dry Friction Force

Studies have shown that normal pressure is a very critical parameter in the design of dry friction dampers [26,27]. According to Figures 1 and 3, the normal pressure of the under-platform damper consists of the normal component of the convective inertia force (centrifugal force), the Coriolis inertial force, and the gravity of damper. The Coriolis inertial force is generated by the movement of the damper in the $y$ direction relative to the bladed disc and varies with the relative velocity to the disc and the rotational speed of the disc. In addition, the component of the gravity of the damper in the normal pressure's direction changes with the rotation of the bladed disc, and it can be ignored, as it is small enough to be, relative to the centrifugal force and the Coriolis inertial force.

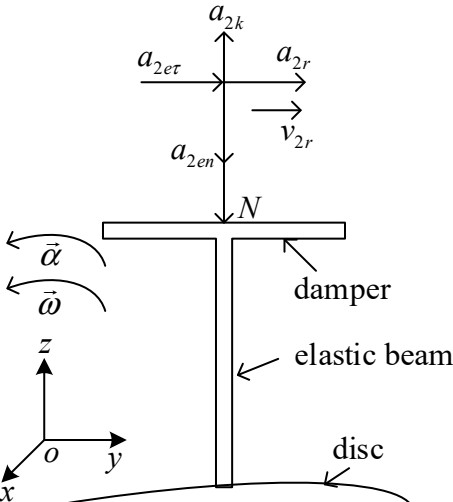

**Figure 3.** Acceleration synthesis and the normal load diagram of the damper.

In Figure 3, the motion of the damper relative to the bladed disc can be approximated as a linear motion along the $y$ direction. The relative velocity, relative acceleration and Coriolis acceleration of each point on the damper are equal, and the direction of Coriolis acceleration changes with the direction of relative velocity. When calculating the total normal pressure, the mass-center acceleration is used to approximately replace the acceleration of each point on the damper.

### 2.3.1. Normal Pressure

The mass of the damper is $m_3$, less than $m_2$ which is the total equivalent mass of the damper and the elastic beam. Calculating the normal pressure $N$, the projection of the damper gravity in the $z$ direction is very small in comparison with the convective inertia force and Coriolis inertia force, thus it can be ignored. As can be seen in Figure 3:

$$m_3(a_{2en} - a_{2k}) = N \tag{5}$$

where

$$\begin{cases} a_{2en} = l_2\omega^2 \\ a_{2k} = 2\omega v_{2r} \end{cases} \tag{6}$$

In Equation (5), $a_{2k}$ and $a_{2en}$ are the Coriolis acceleration and the normal component of the convective acceleration of the damper, respectively. Obviously, normal pressure is caused by the normal component of the convective inertial force (centrifugal force) and the Coriolis inertial force of the damper. The Coriolis inertial force being equal to $-ma_{2k}$ exists with consideration of the bladed disc's rotation. According to Equation (6), the centrifugal force $m_3l_2\omega^2$ contributes positively to the normal pressure, while the direction of relative velocity $v_{2r}$ determines the positive or negative contributions of the Coriolis inertial force to the normal pressure.

### 2.3.2. Dry Friction Force

The dry friction force between two moving bodies is calculated by a bilinear hysteresis model with which stick-slip-separation transition is considered and captured by the bisection method.

As shown in Figure 4, the contact of two moving bodies is considered in this study. The friction force is simulated by a spring which has no initial length and can yield. The contact stiffness is $k_d$, $N$ is the normal pressure, and $\mu$ is the coefficient of kinetic friction. Point 1 is attached to the platform and remains attached at all times, while point 2 is attached to the damper and remains attached at all times. Point $b$ is the sliding contact point, which is initially attached to point 2 with a limiting friction force $\mu N$. Initially the sliding contact point $b$ coincides with point 1 and point 2. The displacements of the platform and the damper relative to the bladed disc are $y_1$ and $y_2$, respectively; the displacement of the sliding contact $b$ relative to the bladed disc is $y_b$. With two bodies moving, point $b$ keeps static with the damper when $|y_1 - y_b|$ is less than $\mu N/k_d$; otherwise, point $b$ keeps static with point 1 and the distance of the two points equals to $\mu N/k_d$.

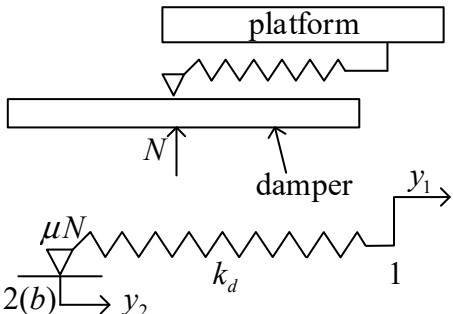

**Figure 4.** Friction contact model between the damper and the platform.

Corresponding to the two local coordinate systems of $y_1$ and $y_2$, assuming that $y_1 = y_2 = y_b = 0$ in the initial state of the system, then the friction force can be determined by Equation (7).

$$f_1 = k_d(y_1 - y_b) \tag{7}$$

Obviously, the friction force is positive when $y_1 \geq y_b$; otherwise, it is negative.

## 3. Numerical Simulation

Referring to the engineering, the angular acceleration of the bladed disc increases from 0, and then decreases to 0 after it reaches the top value. The fourth-order Runge–Kutta algorithm was used to compute the relative vibration responses and study the influence of bladed disc's rotation on the dynamic characteristic of the system. The vibration reduction effect is illustrated. The smooth function is used to describe the angular acceleration as $\alpha(t) = \omega_0[\cos(2\pi t/3 + \pi) + 1]/3$. The working angular velocity of the bladed disc is $\omega_0$, and the angular acceleration $\alpha$ decreases to 0 after the angular velocity reaches $\omega_0$; then, the bladed disc rotates at a uniform angular velocity $\omega_0$. The steady excitation frequency is $f_e$, which equals $\omega_0/2\pi$. $F_0$ is the external excitation amplitude. This paper is a mechanistic study. The parameters of the system can be taken from Table 1. The other parameters are given in the following simulations.

**Table 1.** The parameters of the system.

| Parameters | Values | Parameters | Values |
|---|---|---|---|
| $m_1$ | 0.5 kg | $c_1$ | 1 N s/m |
| $k_1$ | $4 \times 10^5$ N/m | $c_2$ | 1 N s/m |
| $k_d$ | $1 \times 10^6$ N/m | $l_1$ | 0.01 m |
| $\omega_0$ | 600 rad/s | $l_2$ | 0.01 m |
| $\mu$ | 0.2 | $Q_1$ | $F_0 \sin(\omega(t) \times t)$ |

### 3.1. The Analysis of the Vibration Response's Characteristics

1. The parameters are taken from Table 2. Figures 5 and 6 show the simulation results.

**Table 2.** The parameters of the system.

| Parameters | Values |
|---|---|
| $m_2$ | 0.049 kg |
| $m_3$ | 0.04 kg |
| $k_2$ | $6 \times 10^5$ N/m |
| $F_0$ | 400 N |

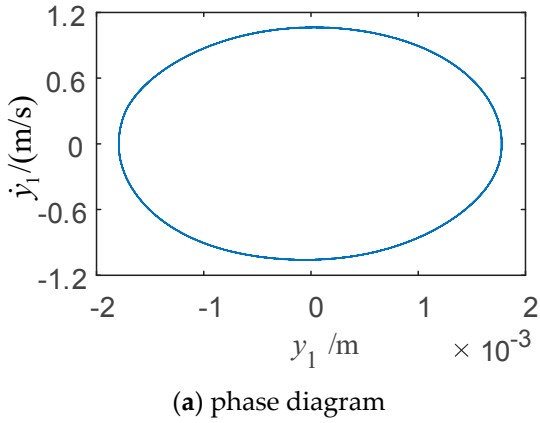

(**a**) phase diagram

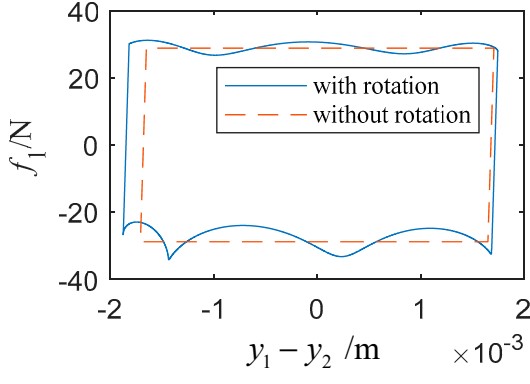

(**b**) hysteretic constructive relation

**Figure 5.** Phase diagram (**a**) and the hysteretic constructive relationship of $f_1$ and $y_1 - y_2$ (**b**) when the system reaches steady-state.

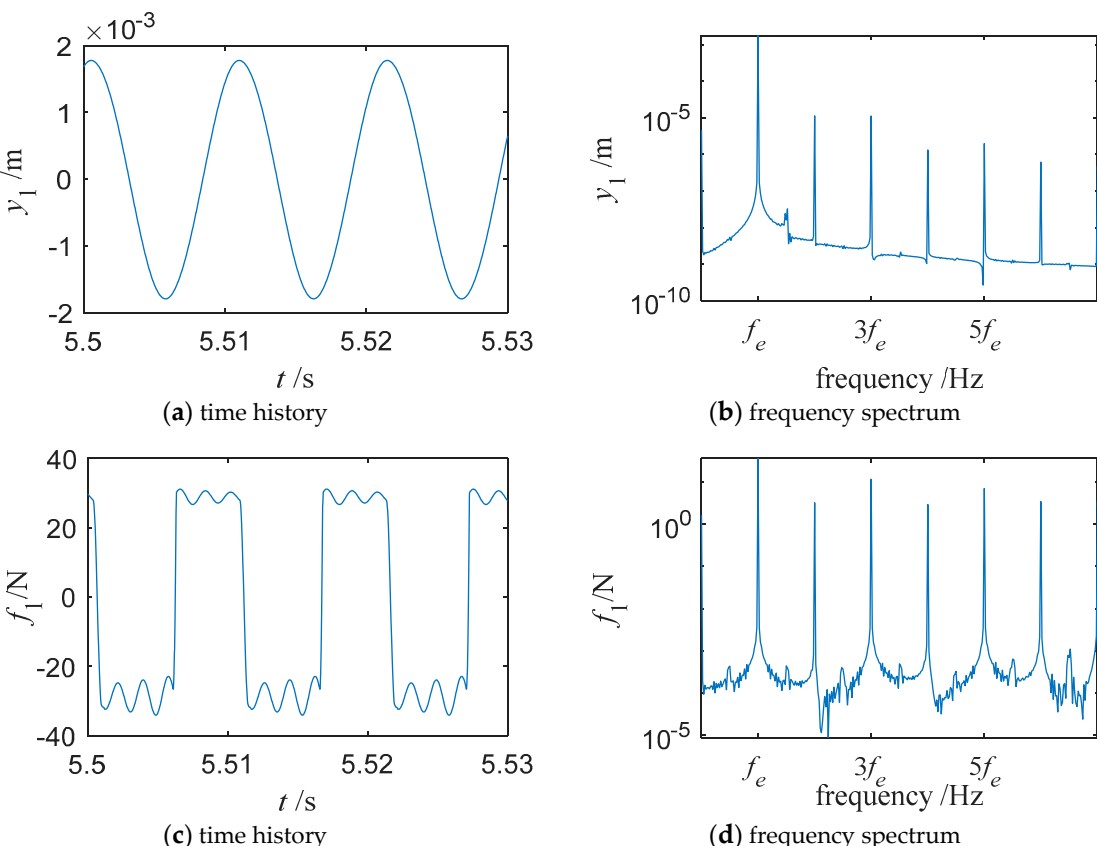

**Figure 6.** Steady-state dynamic characteristics of the response (**a**,**b**) and dry friction force (**c**,**d**).

2. The parameters are taken from Table 3. Figures 7 and 8 show the simulation results.

**Table 3.** The parameters of the system.

| Parameters | Values |
|---|---|
| $m_2$ | 0.079 kg |
| $m_3$ | 0.07 kg |
| $k_2$ | $8 \times 10^5$ N/m |
| $F_0$ | 600 N |

**Figure 7.** Phase diagram (**a**) and the hysteretic constructive relationship of $f_1$ and $y_1 - y_2$ (**b**) when the system reaches steady-state.

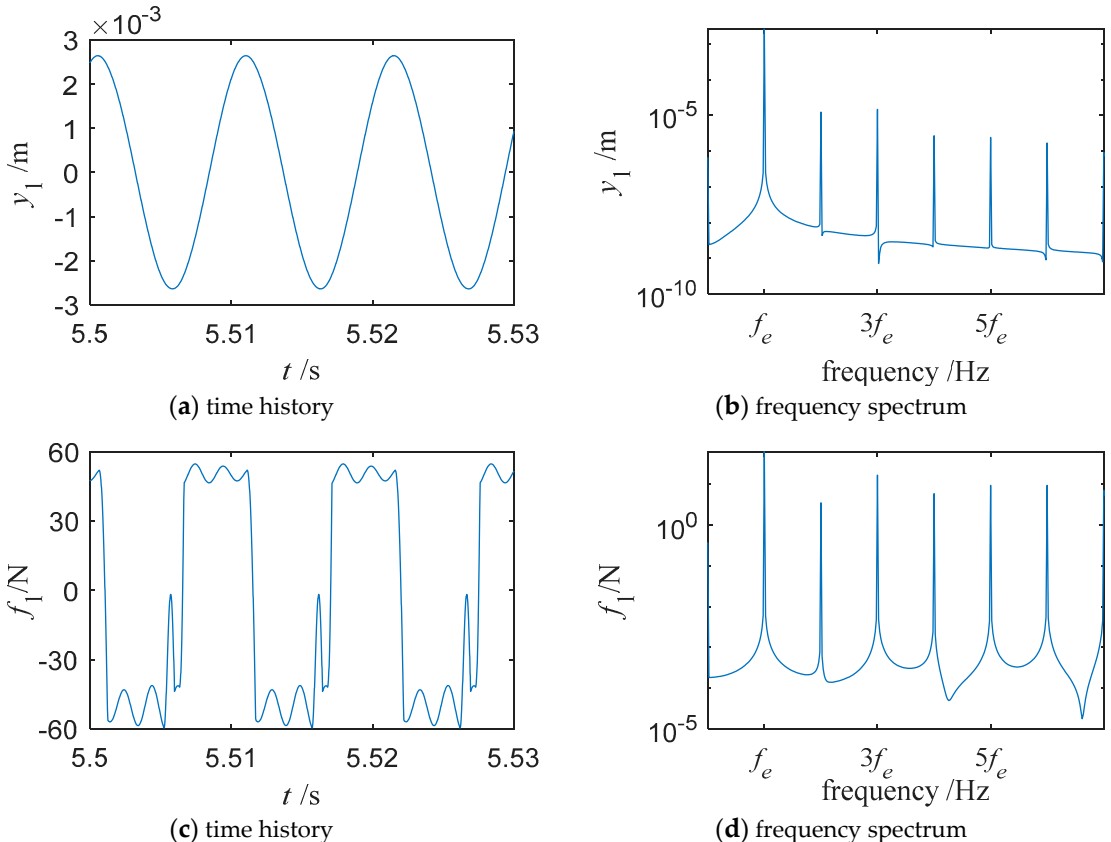

**Figure 8.** Steady-state dynamic characteristics of the response (**a**,**b**) and dry friction force (**c**,**d**).

The two simulations above are typical among the simulating results in this study. When the blade is at steady-state, the motion is periodic from Figures 5a and 7a. Figure 5b and b show the comparison of hysteretic constructive relation of friction force and relative displacement with and without the Coriolis inertial force, and the two constructive relations in the same figure are obviously different. Considering the bladed disc's rotation, the Coriolis inertial force exists and changes the normal pressure; therefore, the hysteresis loop is not symmetrical. As the normal pressure is a very critical parameter in the dry friction damper's design, the dynamic characteristics of the system will be different with that without considering the Coriolis inertial force. In Figures 6 and 8, $f_e$ is the steady excitation frequency, and only odd multiple frequencies of $y_1$ and $f_1$ can be observed. When the mass of the damper, the vibration stiffness of the damper, and the amplitude of external excitation change, there are no fractional frequencies, nor any bifurcation or chaos with the friction contact surface not being separated. The motion of the system is periodic, and the minimum period of the steady-state response $T$ is equal to that of the external excitation, $T = 2\pi/\omega_0$.

### 3.2. The Decision of Steady-State of the Blade

To supply more reference to dry friction platform damper design in engineering, the blade's vibrational reduction of the steady-state response and the transient response will be studied in the next section; therefore, it is necessary to get the moment $t_0$ when the system reaches steady-state. In this section, a method for deciding the stable state of the system is proposed: combining the normalized cross-correlation function (NCCF) and the bisection method. The principle of this method is as followings: choosing steady-state response of the last period as a reference sequence, and a response before the last period as the target sequence. Window size $h$ is the length of target sequence which is taken out each time. The correlation coefficient maximum $C$ of reference sequence and target sequence is calculated by the NCCF. The closer that the value of $C$ is to 1, the more that target sequence is in

agreement with reference sequence. *flag* is a given parameter, and when *C* is greater than its value the response can be considered steady-state. The step length *s* is the moving length of a target sequence each time, and is changeable via the bisection method, which is used to improve the calculation's accuracy and efficiency. Comparing target sequence from back to front with the reference sequence, the moment $t_0$ can be obtained when the step length equals to 1. A computational scheme of the mothed is shown in Figure 9.

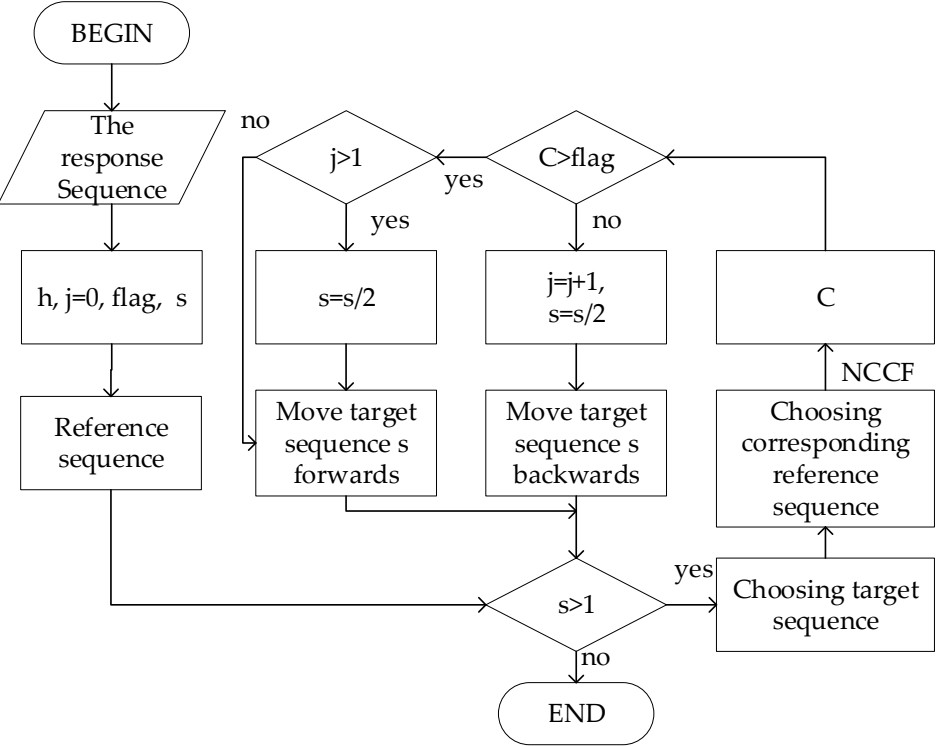

**Figure 9.** The computational scheme of the method.

When *flag* is 0.999, 0.998, and 0.997, the other parameters are shown in Table 2. The results are shown in Figure 10. The difference between the two lines has been amplified by three times for clarity. The moment $t_0$ when the system reaches steady-state was obtained. When *flag* is 0.999, $t_0$ satisfies the accuracy requirement.

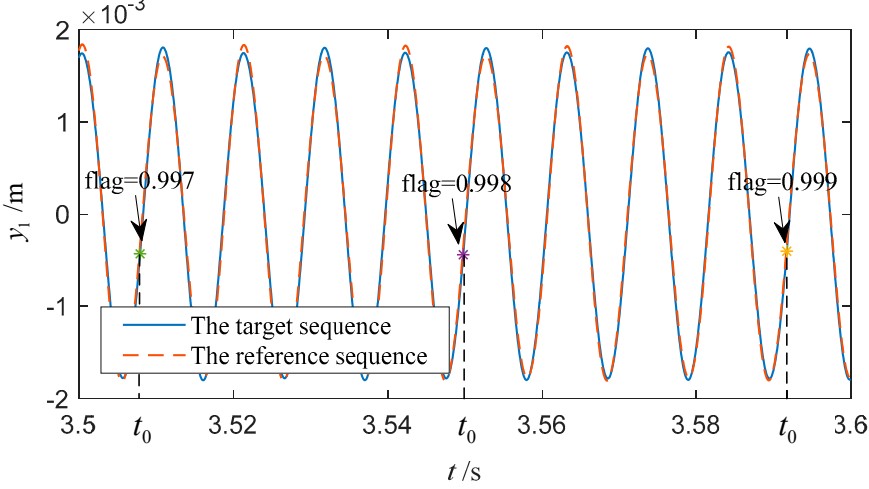

**Figure 10.** The results with different value of flag.

With $t_0$, the response of blade before $t_0$ can be defined as $y_{1t}$, and after $t_0$ can be can be defined as $y_{1s}$; therefore, $y_1$ is divided into $y_{1t}$ and $y_{1s}$.

### 3.3. The Vibration Reduction Characteristics of the System

Based on the analyses done in Sections 3.1 and 3.2, the influence of damper mass, damper vibration stiffness, and external excitation amplitude on the vibration reduction characteristics of the system are studied in this section. Relevant parameters were set as in Table 1. The other parameters are given in the following simulation.

$E_{ds}$ is the vibrational power reduction rate of the steady-state response of the blade. As the steady-state response is periodic, $T$ is the minimum period of the steady-state response, which is equal to the period of the external excitation with the value of $2\pi/\omega_0$. Therefore, $E_{ds}$ can be expressed as Equation (8). $E_{dt}$ is the average power-reduction rate of the transient response of the blade, and can be expressed as Equation (9). The relative displacement of the blade without an under-platform damper is $y'_1$, and the moment when the system without an under-platform damper reaches steady-state is $t'_0$. Similarly, the response of the blade before $t'_0$ can be defined as $y'_{1t}$, and after $t_0$ can be defined as $y'_{1s}$. The response is divided into $y'_{1t}$ and $y'_{1s}$.

$$E_{ds} = \frac{\int_T y'^2_1 dt/T - \int_T y^2_1 dt/T}{\int_T y'^2_1 dt/T} = \frac{\int_T y'^2_1 dt - \int_T y^2_1 dt}{\int_T y'^2_1 dt} = \frac{\int_T (y'^2_1 - y^2_1)dt}{\int_T y'^2_1 dt} \tag{8}$$

$$E_{dt} = \frac{\int_0^{t_0'} y'^2_1 dt/t_0' - \int_0^{t_0} y^2_1 dt/t_0}{\int_0^{t_0'} y'^2_1 dt/t_0'} = \frac{\int_0^{t_0'} y'^2_1 dt - \int_0^{t_0'} t_0' y^2_1 dt/t_0}{\int_0^{t_0'} y'^2_1 dt} = \frac{\int_0^{t_0'} (y'^2_1 - t_0' y^2_1/t_0)dt}{\int_0^{t_0'} y'^2_1 dt} \tag{9}$$

The maximum values of $|y_{1t}|$ and $|y_{1s}|$ are $y_{1tm}$ and $y_{1sm}$ respectively, Similarly, the maximum values of $|y'_{1t}|$ and $|y'_{1s}|$ are $y'_{1tm}$ and $y'_{1sm}$ respectively. $A_{ds}$ is the reduction rate of $y_{1sm}$ and $A_{dt}$ is the reduction rate of $y_{1tm}$. $A_{dt}$ and $A_{ds}$ are expressed in Equation (10).

$$\begin{cases} A_{ds} = \frac{y'_{1sm} - y_{1sm}}{y'_{1sm}} \\ A_{dt} = \frac{y'_{1tm} - y_{1tm}}{y'_{1tm}} \end{cases} \tag{10}$$

#### 3.3.1. The Effect of Damper Mass on the Vibration Reduction

When the working speed is constant, the damper mass has a great influence on the normal pressure. The numerical simulation parameters are taken from Table 4.

**Table 4.** The parameters of the system.

| Parameters | Values |
|---|---|
| $m_2$ | $m_3 + 0.009$kg |
| $m_3$ | 0.04 kg~0.08 kg |
| $k_2$ | $8 \times 10^5$ N/m |
| $F_0$ | 400 N |

The results are as follows:

In Figure 11, $E_{dt}$ and $E_{ds}$ vary with the increase of the damper mass $m_3$; some peaks of $E_{dt}$ and $E_{ds}$ are extant. There is a significant reduction of vibrational power with proper damper mass adding to the blade. In Figure 12, $A_{dt}$ increases with the damper mass's increase, while $A_{ds}$ fluctuates while the damper mass increases. $y_{1tm}$ and $y_{1sm}$ reduce significantly with the proper damper mass adding to the blade. From Figures 11 and 12, the laws of $E_{ds}$ and $A_{ds}$ varying with $m_3$ are basically the same, while the laws of $E_{dt}$ and $A_{dt}$ are obviously different, as the transient response of the blade is complicated. The maximum values of $E_{ds}$ and $A_{ds}$ are smaller than those of $E_{dt}$ and $A_{dt}$ with the same parameters.

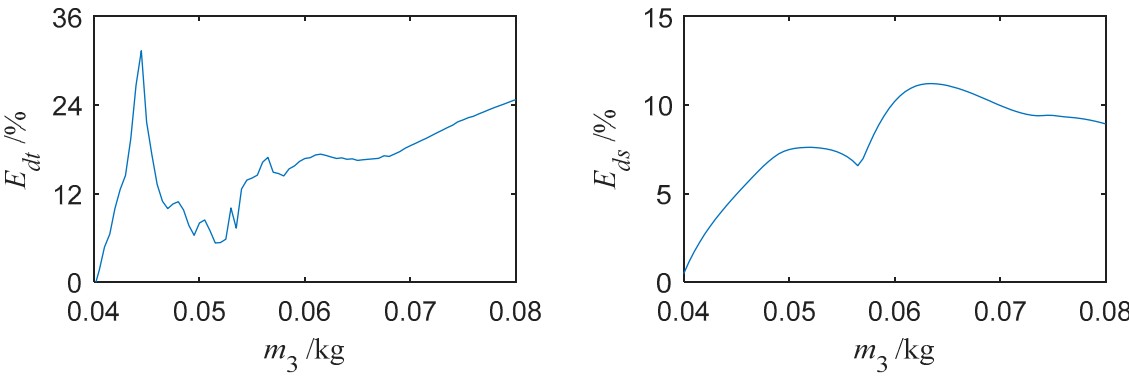

**Figure 11.** The influence of damper mass on the vibrational power reduction.

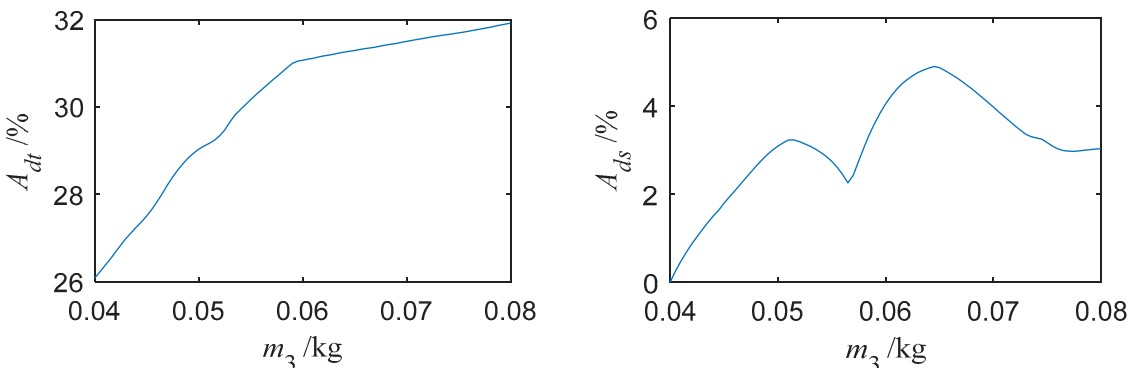

**Figure 12.** The influence of damper mass on the reduction of the maximum absolute value of vibrational response.

### 3.3.2. The Effect of a Damper's Vibrational Stiffness on the Vibration Reduction

The effect of a damper's vibration stiffness on the vibration reduction of the blade with an under-platform damper was studied. The parameter values are shown in Table 5.

**Table 5.** The parameters of the system.

| Parameters | Values |
|:---:|:---:|
| $m_2$ | 0.059 kg |
| $m_3$ | 0.05 kg |
| $k_2$ | $6 \times 10^5$ N/m~$1 \times 10^6$ N/m |
| $F_0$ | 400 N |

The results are as follows:

In Figure 13, $E_{dt}$ and $E_{ds}$ fluctuate with the increasing of the damper stiffness $k_2$, and there is a significant reduction of vibrational power with proper damper stiffness $k_2$. In Figure 14, the damper stiffness $k_2$ has an obvious influence on $A_{dt}$ and $A_{ds}$; $y_{1tm}$ and $y_{1sm}$ reduce significantly with the proper damper stiffness. From Figures 13 and 14, the laws of $E_{ds}$ and $E_{dt}$ varying with $k_2$, are basically the same, while the laws of $E_{ds}$ and $E_{dt}$ are obviously different. The maximum values of $E_{ds}$ and $A_{ds}$ are smaller than those of $E_{dt}$ and $A_{dt}$ with the same parameters.

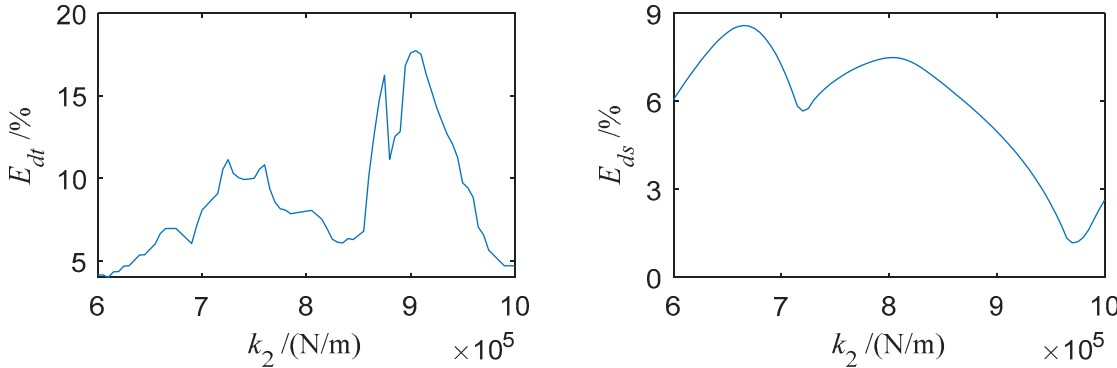

**Figure 13.** The influence of damper stiffness on the vibrational power reduction.

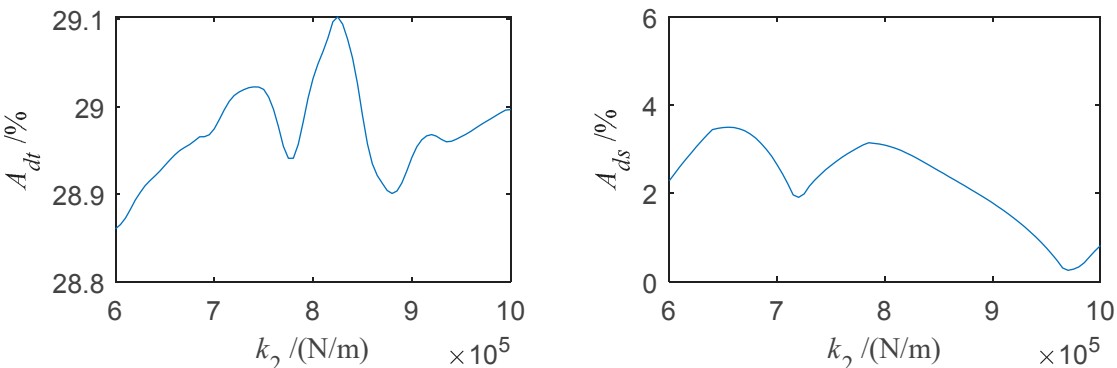

**Figure 14.** The influence of damper stiffness on the reduction of maximum absolute value of vibrational response.

### 3.3.3. The Effect of External Excitation Amplitude on the Vibrational Reduction

The parameters are shown in Table 6. The results are shown in Figures 13 and 14.

**Table 6.** The parameters of the system.

| Parameters | Values |
|:---:|:---:|
| $m_2$ | 0.059 kg |
| $m_3$ | 0.05 kg |
| $k_2$ | $8 \times 10^5$ N/m |
| $F_0$ | 200 N~800 N |

In Figure 15, $E_{dt}$ and $E_{ds}$ basically decrease with the increase of external excitation amplitude $F_0$. In Figure 16, $A_{dt}$ and $A_{ds}$ decrease with increasing $F_0$. From Figures 15 and 16, the increasing external excitation amplitude causes the vibrational reduction effect of the blade to decrease, obviously, and a larger normal pressure would be needed to make the damper work well. The laws of $E_{ds}$ and $E_{dt}$ varying with $k_2$, are basically the same, as are $E_{ds}$ and $E_{dt}$. Besides, $E_{dt}$ could be negative with $F_0$ increasing, which needs to be considered when engineering damper designs.

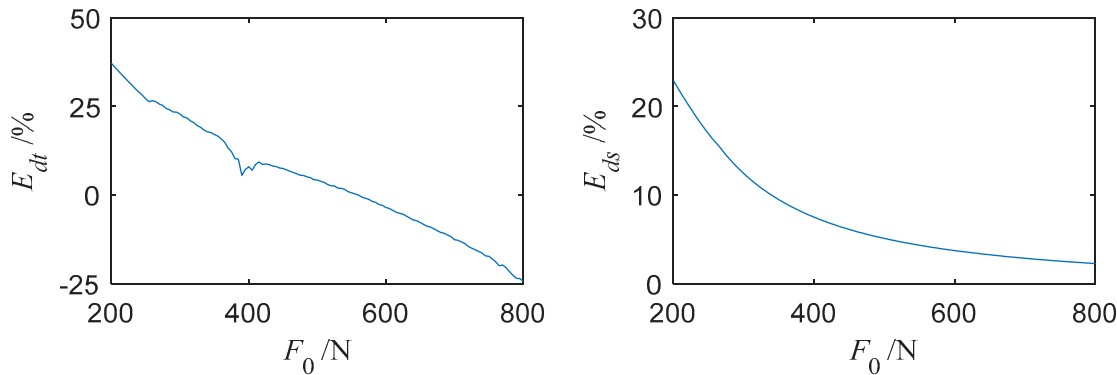

**Figure 15.** The influence of external excitation amplitude on the vibrational power reduction.

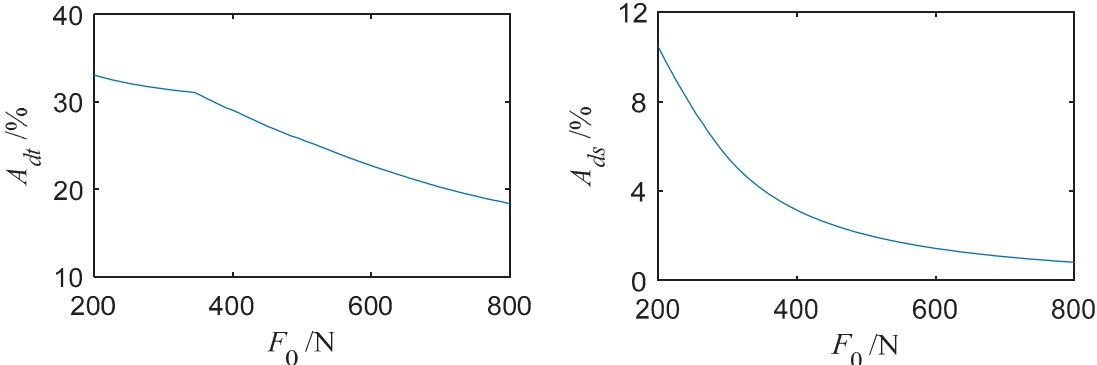

**Figure 16.** The influence of external excitation amplitude on the reduction of the maximum absolute value of vibrational response.

## 4. Conclusions

The paper presents a study on the dynamic characteristics of a blade's vibration response relative to a bladed disc by considering the bladed disc's rotation, and an effective method for deciding the steady-state of the blade is proposed. The influences of the mass of the damper, the damper's vibrational stiffness, and the external excitation amplitude are discussed. The following conclusions can be drawn:

1. The dynamic model and analysis method presented in this paper are effective to study the influence of a bladed disc's rotation on the dynamic characteristics of a turbine blade. The changes of the convective inertial force and Coriolis inertial force during the rotation of the bladed disc have a significant influence on the hysteretic constructive relationship of friction-relative displacement and the characteristics of the system dynamics.
2. When the friction contact surface is not detached, changing the damper mass, the damper vibration stiffness, and the external excitation amplitude, results in only higher harmonics in the system response and friction force, and in the system's response, bifurcation and chaos cannot be observed.
3. With proper parameters, adding a platform damper will make the turbine blade's vibration reduce obviously. When the steady-state response is periodic, the reduction law of the average power of blade vibration and the maximum absolute value of the steady-state response are basically the same. However, reduction laws of the average power of blade vibration and the maximum absolute value of the transient response are different. With some parameters, the reduction effect of the transient response may be negative. Greater normal pressure would be needed to keep the damper working well when the external excitation amplitude increases.
4. In engineering, the vibrational reduction effects of the steady-state response and the transient response should be analyzed comprehensively in the under platform-damper design.

**Author Contributions:** S.H. conceived the paper. S.H., W.J., and Z.Y. wrote the paper and clarified the key methods and results. B.H. helped in the programming and data analysis. J.Z. edited the manuscript.

**Funding:** This research was funded by the National Key R&D Program of China (grant number 2016YFE0125600) and the National Natural Science Foundation of China (grant number 51405452).

**Acknowledgments:** The authors are grateful to the discussion on compositive motion with Doctor Xu Zhang.

**Conflicts of Interest:** The authors declare no conflict of interest.

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
