# Peer review of "Dynamics of a Turbine Blade with an Under-Platform Damper Considering the Bladed Disc’s Rotation"

_applsci, doi:10.3390/app9194181_

Round 1

Reviewer 1 Report

Title: Dynamics of the Turbine Blade with Under-Platform Damper Considering
Bladed Disc Rotating.

This paper presents the dynamic model of the under-platform damper considering the bladed disc rotating a transient state and steady state. It is a topic of interest to the researchers in the related areas but the title and introduction section may have misleading and the paper needs very significant improvement before acceptance for publication.

My detailed comments are as follows:

Please go through the entire article one more time and revise. There are some problems, such as syntax and sentence construction that need to be corrected. For examples: the improper use of “researches”; the mixed use of “bladed disc” and “blade disc”, please only use one of them; “To determine to what…” in page 2 line 63; long sentence “when the bladed disc…approximate calculation” in page 3, line 117-120, “and” in line 119 should be deleted; long sentence “in general… and chaos” in page 9 line 235-237; the tense in the introduction section should be checked, such as page 2 line 64 “is”, line 76”are”; and et al. Please use subsections or delete “researches about …:” at the paragraph heads in the introduction section. The dot point performance is not suitable for a scientific paper writing. Similarly, in Section 3.1, please do not use the number point direct in the text. Table or subsection should be better. Should that the under-platform damper be sheared between two blades? If do so,  figure 2 and the dynamic model should have some changes. Page 3, line 116 (the damper is ) “slidable along the height of the elastic beam without friction” , is that mean the damper is free motion in the radial direction of the disc (z direction in figure 1)? If do so, the normal pressure would has problem. What is N1 in Eq.(1)? Page 6, line 177, m3 is less than m2. Why? Please explain more on their definition, or difference. Figure 4 is not will explained and need improvement. Is that “b” a contact point, “1” and “2” are location point?  Also y2 is not shown in the figure 4. Page 7 line 207, what is the “engine”? Page 7 line 215, the excitation frequency fe is not well explained. Is there a relation betweent he frequency and the Q1 equation just before? Figure 10, what are the parameters for the simulation result? Are they the same as one of the two sets in section 3.1? Page 10 line 264-265, please check the font of the text. Page 11 line 275 “Eds” should be “Edt” Page 11 line 276-277, is the “dry friction platform damper” the same as “under-platform damper”? If do so, please use the same word or short the sentence. It may confuse the reader. In this paper, the under-platform damper is the modelled by considering the rotating condition of the bladed disc. But how the authors verify the accuracy of the model? There are no experiments or other ways making comparison with the model in this paper. According to the title, abstract and the introduction, one novelty of the paper should be the consideration of bladed disc rotating. However, through over the paper, only transient responses are discussed. Thoroughness of the analysis can be deeper, such as the eccentric motion of the disc and so on. The novelty can be improved.

I think this manuscript should be reconsidered after major revision.

Reviewer 2 Report

See attached file

Round 2

Reviewer 1 Report

I would like to thank the authors for their time and valuable responses, all enquires have been corrected. But I would also recommend to extend the area of your work by considering improve the thoroughness of the analysis, such as the eccentric motion of the disc and so on. The manuscript is acceptable to publish.